# Influence of *Debaryomyces hansenii* on bacterial lactase gene diversity in intestinal mucosa of mice with antibiotic-associated diarrhea

**Yunshan He[1], Yuan Tang[1], Maijiao Peng[1], Guozhen Xie[1], Wenge Li[2]\*, Zhoujin Tan[1]\***

**1** Hunan University of Chinese Medicine, Changsha, Hunan Province, China, **2** Hunan Institute of Nuclear Agricultural Sciences and Space-induced Breeding, Changsha, Hunan province, China

\* 641386565@qq.com (WL); tanzhjin@sohu.com (ZT)

## Abstract

### Aim

The current study aimed to investigate the effects of *Debaryomyces hansenii* on the diversity of bacterial lactase gene in the intestinal mucosa of antibiotic-associated diarrhea (AAD) mice.

### Methods

Eighteen mice were randomly divided into three groups (6 mice per group): healthy control group, diarrhea model group and *D. hansenii* treatment group. The antibiotic-associated diarrhea model was established by intragastric administration with a mixture of cephradine and gentamicin sulfate (23.33 mL·kg$^{-1}$·d$^{-1}$) twice a day for 5 days continuously. After establishing the AAD model, the mice in the *D. hansenii* treatment group were gavaged with *D. hansenii* for three days, while other groups were gavaged with distilled water. Then, the intestinal mucosa of all three groups was collected and DNA was extracted in an aseptic environment for the following analysis.

### Results

The difference in the richness and homogeneity of the bacterial lactase gene among all samples were inapparent, as the difference in the Chao1, ACE, Simpson and Shannon indices among the three groups were insignificant (*P*>0.05). NMDS analysis also showed that the distance of the samples among the three groups was unobvious. Furthermore, the bacterial lactase gene in the mucosa mainly originated from *Actinobacteria*, *Firmicutes* and *Proteobacteria*. Compared with the healthy control group, the abundance of lactase genes originating from *Cupriavidus*, *Lysobacter*, *Citrobacter*, *Enterobacter* and *Pseudomonas* was increased in the *D. hansenii* treatment group, while the lactase gene from *Acidovorax* and *Stenotrophomonas* decreased (*p* < 0.01 or *p* < 0.05) in the diarrhea model group and the *D. hansenii* treatment group.

**Data Availability Statement:** All relevant data are within the manuscript.

**Funding:** The research was supported by the National Natural Science Foundation of China (No.81573951) to ZT. The funder had no role in study design, data collection and analysis, decision to publish, or preparation of the manuscript.

**Competing interests:** The authors have declared that no competing interests exist.

## Conclusion

*D. hansenii* was capable of improving the growth of some key lactase-producing bacteria like *Deinococcus*, *Cupriavidus* and *Lysobacter* for treating AAD.

## Introduction

Lactase, also known as β-galactosidase, is an important enzyme for intestinal function, which can hydrolyze lactose into glucose and galactose, allowing lactose to be absorbed and utilized by human beings and animals. Because of inactivity and deficiency of lactase, lactose is fermented by intestinal bacteria to produce significant amounts of short-chain fatty acid (SCFA) and hydrogen rather than hydrolyzed, which cause diarrhea and is referred to as lactose intolerance (LI) [1,2]. Usually, intestinal lactase is produced by intestinal epithelial cells, microorganisms and obtained from exogenous lactase[3]. Nevertheless, with the widespread use of antibiotics in the clinic, increasing attention has been paid to antibiotic-associated diarrhea (AAD). Currently, it is believed that AAD, persistent diarrhea and lactase-deficiency diarrhea are all related to low lactase activity, which is probably due to a reduced number of lactases, affecting the arrangement and shedding of villi[4]. Fortunately, most diarrhea can be relieved by oral probiotics such as *Lactobacillus*, *Bifidobacterium*, *Saccharomyces boulardii* and by lactase supplementation [5–8].

Approximately 1000 species of bacteria inhabit the human intestinal tract, which plays an important role in metabolism, immunity and other physiological functions[9,10]. The intestinal microbes that inhabit the enteric cavity and epithelial cells of the mucosa establish a complete intestinal mucosal barrier together with intestinal epithelial cells, mucus and secrete so as to play a critical role in maintaining the homeostasis and healthy[11,12]. However, the extensive use of antibiotics pose a risk of improper application, which could cause an imbalance in the intestinal microbiota, destruction of the mucosal barrier related diseases and eventually [13]. Our previous research reported that antibiotics decrease the diversity of bacterial lactase genes in the intestinal contents, and they transforme their community structures[14].

Yeast has been widely used in medicine, food, beverage, alcohol and other industries because of its rapid propagation ability and efficient metabolism[15]. *D. hansenii*, isolated from the natural environment, food or intestinal tract, is one of the most important unconventional yeasts. A large number of studies verified that *D. hansenii* plays an important role in the fermentation of cheese and sausage and the production of fuel alcohol. It can produce critical esters in sausage manufacture, as well as the thermophilic lactase necessary to produce fuel alcohol[16–20]. In addition, it can also produce antitoxins to inhibit the growth of the harmful bacteria such as *Candida* [21]. Our previous studies showed that *D. hansenii*, which was isolated from the mouse intestine[19], was able to tolerate a high acid and high bile salt environment, and it had high viability in the artificial gastrointestinal fluid environment. In addition, the combination of 25% *D. hansenii* and 25% Qiweibaizhusan was used to modulate the population of total intestinal bacteria and *Escherichia coli*, and it can also restore the bacterial diversity in mice with dysfunctional diarrhea[22–25]. The vast majority of previous studies focused on the pathogenesis or treatment of AAD, and our previous studies reported the influence of *D. hansenii* on AAD based on intestinal bacterial diversity as well. The current research aimed to investigate the effect of *D. hansenii* on AAD based on the diversity of the bacterial lactase gene in intestinal mucosa by using high-throughput sequencing. It can provide further experimental basis for the development and utilization of *D. hansenii* as a new microecological preparation.

## Materials and methods

### Reagents and medicine

Cephradine capsules (batch number: 151101) were purchased from Suzhou Chung-Hwa Chemical & Pharmaceutical Industrial Co. Ltd., and gentamicin sulfate (batch number: 5150307) was purchased from Yichang Pharmaceutical Industrial Co. Ltd. The two antibiotics were then prepared into a mixture at a concentration of 62.5 g·L$^{-1}$[26]. The DNA extraction reagents (such as Proteinase K, lysozyme, Tris-saturated phenol-chloroform-isoamyl alcohol (25:24:1) and TE buffer) were purchased from Beijing Dingguo Changsheng Biotechnology Co. Ltd., and prepared in the lab (such as 10% SDS, chloroform-isoamyl alcohol (24:1), 5 mol·L$^{-1}$ NaCl, 0.1 mol·L$^{-1}$ PBS and CTAB/NaCl). *D. hansenii*, which was provided by the laboratory and shaken at 28˚C for 36 hours after being inoculated into liquid Potato Sucrose medium in a 300 mL erlenmeyer flask. The cells were then gathered by centrifugation at 2000×$g$ for 4 minutes after washed 1∼2 times repeatedly with sterile stroke-physiological saline solution. The above cells were diluted to 10$^{10}$ mL$^{-1}$ with sterile storke-physiological saline solution eventually after being counted by hemocytometer, and stored at 4˚C for subsequent experiments[27].

### Animals and procedures

Eighteen specific pathogen-free (SPF) Kunming (KM) mice (nine male and nine female, one month old), weighting 20±2 g, were purchased from Hunan Slaccas Jingda Laboratory Animal Company with license number SCXK (Xiang) 2013–0004. All procedures involving animals were performed according to protocols approved by the Institutional Animal Care and Use Committee of Hunan University of Chinese Medicine. Mice were randomly divided into three groups (6 mice per group, half male and half female): healthy control group (tlcm), diarrhea model group (tlmm) and *D. hansenii* treatment group (tljm). Mice in the healthy control group were gavaged with 0.35 mL of sterile water twice a day for 5 days. To induce diarrhea, mice in both the diarrhea model group and the *D. hansenii* treatment group were injected intragastrically with a mixture of gentamicin sulfate and cefradine capsules (23.33 mL·kg$^{-1}$·d$^{-1}$) twice a day for 5 days by following the procedures described previously[28]. After diarrhea symptoms appeared (such as erected coat, reduced intake, watery stool and declined activity), the mice in the *D. hansenii* treatment group were treated with *D. hansenii* by intragastric administration (0.35 mL, 10$^{10}$ mL$^{-1}$), and the other two groups were given aseptic water twice a day for 3 days. Then, the mucosa from the jejunum to ileum were scraped with a cover slip after the contents were extruded and the intestine was washed twice with sterile saline solution in a germ-free environment (each sample contains mucosa from two mice: one male and the other female, which has three samples per group), and then were frozen immediately and stored at -20˚C for further use [29].

Ethical approval was obtained from the Animal Ethics and Welfare Committee of Hunan University of Chinese Medicine.

### Total DNA extraction from intestinal mucosa

Total DNA from the intestinal mucosa was extracted following to our previous reports[30,31]. A total of 2.0 g of mucosa was weighed in a germ-free environment and preprocessed with 0.1 mol·L$^{-1}$ phosphate buffer solution (PBS) and acetone. The cells were collected by the above method and resuspended in 4 mL of TE buffer. Cells walls were disrupted by lysozyme, and total DNA was purified and extracted by proteinase K, CTAB/NaCl, Tris saturated phenol-chloroform-isoamyl alcohol (25:24:1), chloroform-isoamyl alcohol (24:1), absolute ethyl

alcohol and sodium acetate after cells wall-broken by lysozyme. Eventually the DNA was dissolved in 50 μL of TE buffer for further analysis.

## PCR amplification of the mucosal bacterial lactase gene and high-throughput sequencing

The universal primers for PCR amplification were designed according to the lactase gene sequences of *Lactobacillus* and *Escherichia coli* from NCBI and synthesized by Shanghai Personal Biotechnology Co., Ltd.[32]. The forward primer was 5'-TRRGCAACGAATACGGSTG-3', and the reserve primer was 5'-ACCATGAARTTSGTGGTSARCGG-3'. After PCR mixtures were prepared (25 μL) (including 5 μL of 5 × high GC buffer, 0.25 μL of Q5 high-fidelity DNA polymerase, 1 μL of 10 μmol·L$^{-1}$ forward primer, 0.5 μL of 10 mmol·L$^{-1}$ dNTP, 1 μL of 10 μmol·L$^{-1}$ reserve primer, 1 μL of DNA template and 11.25 μL of sterilized ddH$_2$O) and added to 0.5 mL PCR tubes, the amplification was carried out as follows: conditions: initial denaturation at 98˚C for 30 s, 32 cycles of denaturation at 98˚C for 15 s, annealing at 46˚C for 30 s and extension at 72˚C for 30 s, then extension at 72˚C for 5 min and holding at 4˚C. PCR products of the bacterial lactase gene were purified and then detected using high-throughput sequencing, which was performed by Shanghai Personal Biotechnology Co., Ltd.

## Gene diversity and statistical analysis

The software available online, including QIIME (http://qiime.org/)[33] and Mothur (http://www.mothur.org/), were used to analyze the sequencing results. Alpha diversity analysis, including Chao1, ACE, Simpson and Shannon indices, was applied to identify the richness and uniformity of the intestinal mucosa bacterial lactase gene by determining the operational taxonomic units (OTUs)[34–37]. Principle component analysis (PCA)[36] was used to analyze the community difference of lactase-producing bacteria according to the distance among individuals. The source and abundance of the bacterial lactase gene at the specific taxonomic levels are presented in the figure containing species evolution and abundance information and statistics table of relative abundance statistics table. Then, our measurement data were analyzed using the SPSS 21.0 software (IBM Corp, Armonk, NY, USA), of which one-way ANOVA was applied to compare the statistical significance of differences, with *p* value.

## Results

### Sequence statistics and OTU analysis

The diversity and richness of lactase gene can be well studied by measuring and analyzing OTUs (operational taxonomic unit). **Fig 1** shows that the numbers of OTUs of the healthy control group, diarrhea model group and *D. hansenii* treatment group was 298, 435 and 326, respectively. The results showed that 45 OTUs were unique to the healthy control group, 202 OTUs were unique to the diarrhea model group and 57 OTUs were unique to the *D. hansenii* treatment group. These results suggested that antibiotic modeling increased the number of OTUs of the lactase gene from the intestinal mucosal bacteria in mice, while the number of OTUs of antibiotic models can be returned to normal following treatment with *D. hansenii*.

### Alpha diversity of the bacterial lactase gene from the intestinal mucosa of AAD mice treated with *D. hansenii*

By drawing rarefaction curves, the diversity of each sample could be measured to some extent. **Fig 2** shows that each curve tended to flatten with the increase in the number of measured

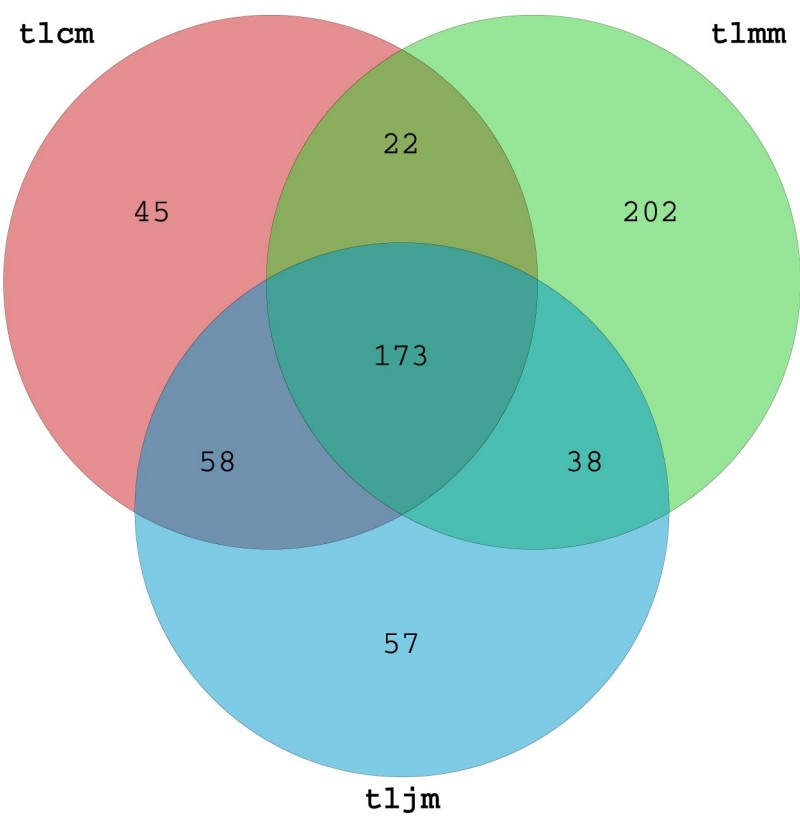

**Fig 1. OTUs of the bacterial lactase genes from intestinal mucosa.** tlcm, healthy control group; tlmm, diarrhea model group; tljm. *D. hansenii* treatment group.

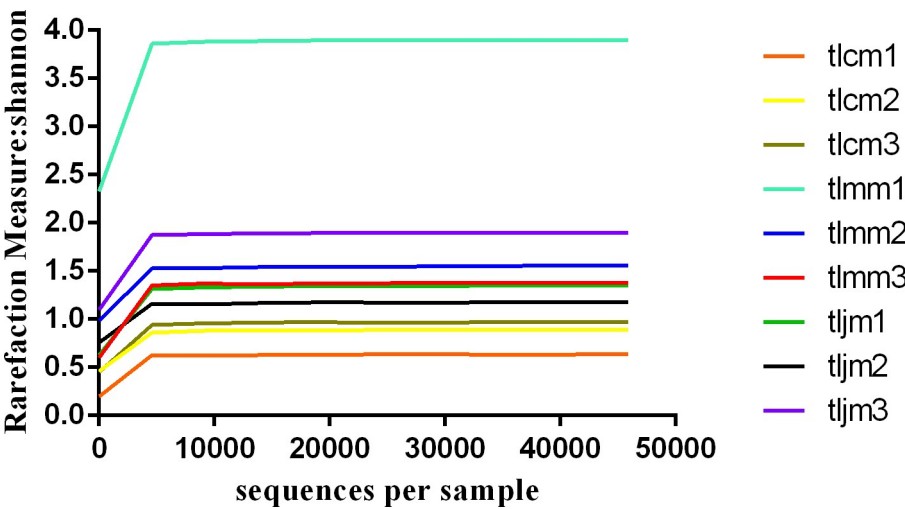

**Fig 2. Rarefaction curve of the bacterial lactase gene from the intestinal mucosa in each sample.** Note: The abscissa represents the number of lactase gene sequences extracted randomly; the ordinate represents the number of observed OTUs of the bacterial lactase gene. The data indicate that the sequencing tended to be saturated and that increasing the amount of data would have no significant effect on obtaining new OTUs when the curve tended to be flat; tlcm1-3, tlmm1-3, tljm1-3 are healthy control group samples 1–3, diarrhea model group 1–3 and *D. hansenii* treatment samples 1–3, respectively.

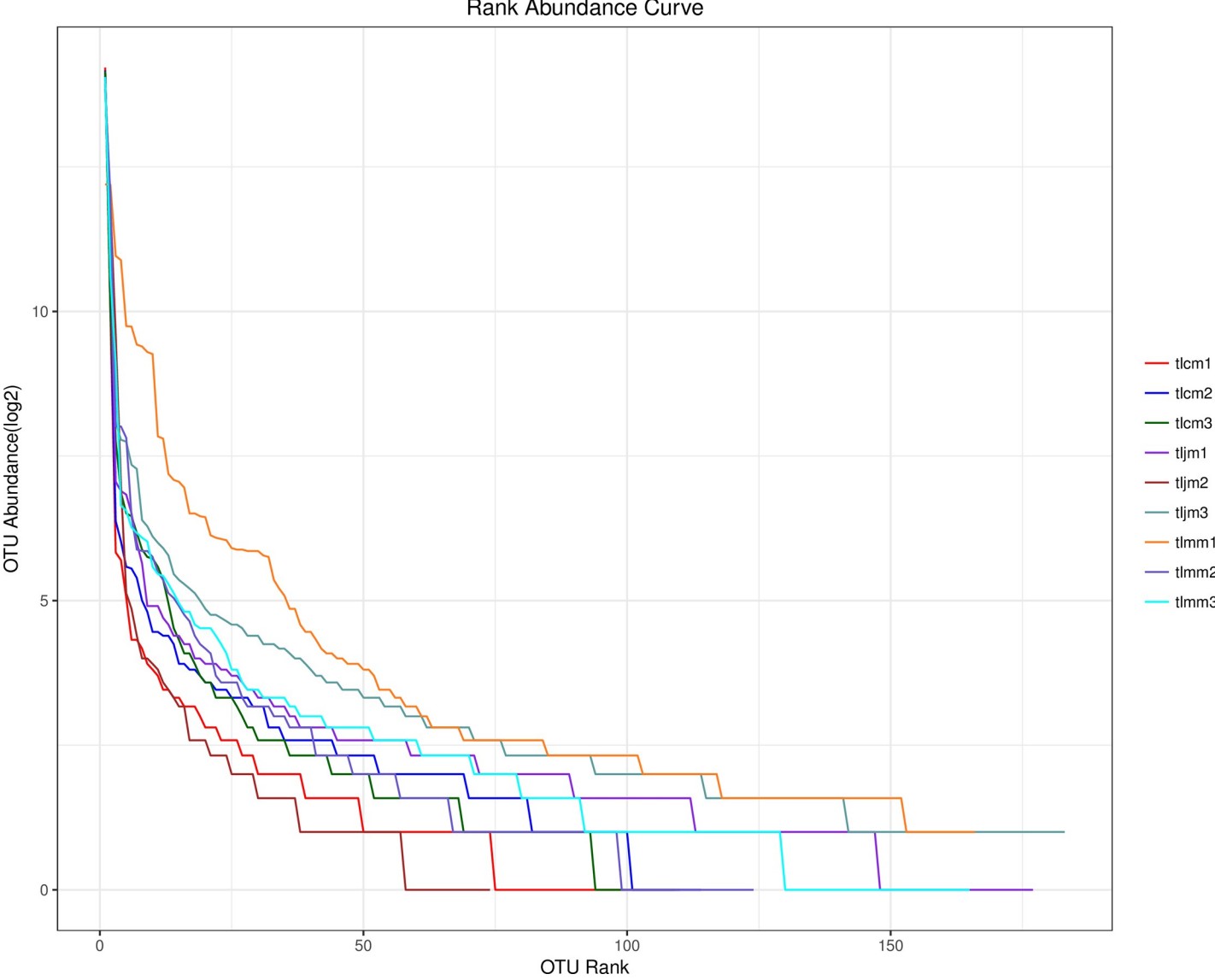

**Fig 3. Rank abundance curve of each sample.** Note: The abscissa represents the ordinal of the OTU, and the ordinate represents the abundance of the OTU. The larger the curve span, the richer the composition of the species was. The flatter the curve, the higher the evenness the species composition was. tlcm 1–3 represented the healthy control group samples 1–3, tlmm1-3 represented the diarrhea model group samples 1–3, tljm 1–3 represented the *D. hansenii* treatment samples 1–3.

sequences. This result suggested that the sequencing results were more than enough to reflect the current sample containing the intestinal mucosa lactase gene diversity.

The rank abundance curve and alpha indices can be used to determine the richness and uniformity of the bacterial lactase gene from intestinal mucosa to evaluate the therapeutic efficacy of *D. hansenii* treatment on AAD. As shown in **Fig 3**, there was no significant difference in the length on abscissa and the gentleness, which indicated that *D. hansenii* treatment had no significant impact on the richness and uniformity of the bacterial lactase gene among the healthy control group and diarrhea model group. From the alpha indices, we determined that there was no significant difference in the Chao1, ACE, Simpson and Shannon indices among the three groups (**Table 1**).

**Table 1. Alpha diversity indices of the bacterial lactase gene from the intestinal mucosa of antibiotic-associated diarrhea mice treated with *D. hansenii*.**

| Group | Chao1 | ACE | Shannon | Simpson |
|-------|-------|-----|---------|---------|
| tlcm | 121.89±8.81 | 127.93±13.58 | 0.80±0.18 | 0.18±0.04 |
| tlmm | 160.33±24.15 | 167.13±13.60 | 2.26±1.43 | 0.54±0.29 |
| tljm | 150.85±61.02 | 154.10±58.53 | 1.45±0.38 | 0.38±0.05 |

**Note:** The larger Chao1 and ACE indices represent higher richness of the bacterial lactase gene. The larger Shannon and Simpson index a represents higher diversity of bacterial lactase genes. tlcm, tlmm and tljm represented the healthy control group, diarrhea model group, and *D. hansenii* treatment group, respectively.

## Beta diversity of bacterial lactase genes in the intestinal mucosa of AAD diarrhea mice treated with *D. hansenii*

PCA(principal component analysis) analysis can extract the most important differences between samples from the original data. As shown in **Fig 4,** that the three samples in the healthy control group and the *D. hansenii* treatment group were relatively close to each other, while the distance between the groups was relatively far, which can be clearly distinguished. The distribution of the three samples in the diarrhea model group was very scattered and not completely separated from the healthy control group and the *D. hansenii* treatment group. This finding suggests that antibiotics altered the structure of bacterial lactase-producing genes in the intestinal mucosa, and the percentages attributed to the variations in PC1 and PC2 were 97.32% and 2.66%, respectively.

The differences in the community were measured by distance comparison among individuals through NMDS analysis. The distribution of samples in the healthy control group was more concentrated than that in the other groups, and the distance between the diarrhea model group and the healthy control group was approximately the same distance as that between the *D. hansenii* treatment group and the healthy control group (**Fig 5**). The results indicated that *D. hansenii* treatment had no significant effect on the recovery of the community structure of the bacterial lactase gene from the intestinal mucosa.

## Abundance and source of the bacterial lactase gene from the intestinal mucosa of AAD mice treated with *D. hansenii*

As shown in **Fig 6(A)**, at the genus level, the number of known lactase-producing bacteria detected in the healthy control group, diarrhea model group and *D. hansenii* treatment group were 16, 12, and 16, respectively. *Mesorhizobium* was only found in the healthy control group, and *Plasmodium* was only detected in the *D. hansenii* treatment group, *Sphingomonas* and *Bordetella* were not detected in the diarrhea model group. In addition, a number of other bacterial lactase genes (other, including some genera with low abundance or unclassified) and some new lactase-producing bacterial genera (no Blast hits) were detected. Compared with the healthy control group, the abundance of *Cupriavidus* was increased significantly after treatment with *D. hansenii*. The lactase genes originating from *Acidovorax* and *Stenotrophomonas* were lower in the diarrhea model group and *D. hansenii* treatment group than the healthy control group. Conversely, the lactase gene originated from *Citrobacter*, *Enterobacter* and *Pseudomonas* in the diarrhea model group and *D. hansenii* treatment group was lower than that in the healthy control group. However, the lactase genes of five genera showed no significant difference between the diarrhea model group and the *D. hansenii* treatment group. Through further analysis with LEfSe, shownin **Fig 6(B)**, it was found that *Sphingomonadaceae*, *Sphingomonas* and *Sphingomonadales* were the key colony members in the *D. hansenii* treatment group, while no key colony members were found in the healthy control group or diarrhea model group.

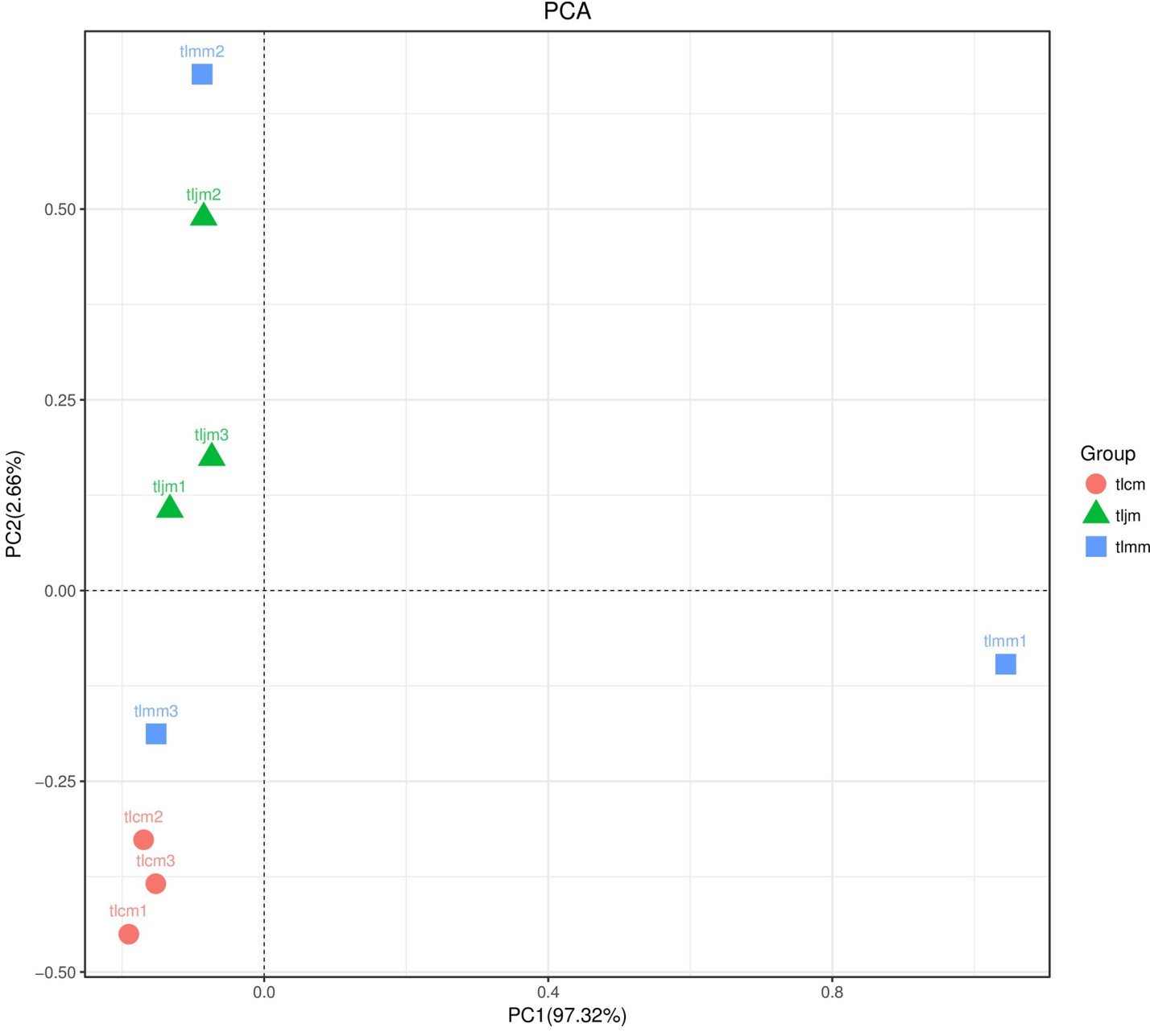

**Fig 4. PCA analysis of the bacterial lactase gene from intestinal mucosa.** Each point represents a sample, and different color points belong to different groups. The closer the two points are, the smaller the difference of the lactase-producing bacterial community structure between the two samples is. tlcm, tlmm and tljm were the healthy control group, diarrhea model group and *D. hansenii* treatment group respectively.

The species evolution and abundance information were used for source and abundance analysis of the bacterial lactase gene at different classification levels (**Fig 7**). The evolution tree showed that the bacterial lactase genes in intestinal mucosa mainly originated from *Actinobacteria*, *Firmicutes* and *Proteobacteria*. The abundance of the lactase gene in *Actinobacteria* from the *D. hansenii* treatment group was the highest, followed by the healthy control group and the diarrhea model group. Lactase genes from *Aeromonas*, *Curvibacter* and *Deinococcus* were only detected in the *D. hansenii* treatment group, while those from *Enterobacteriaceae* and

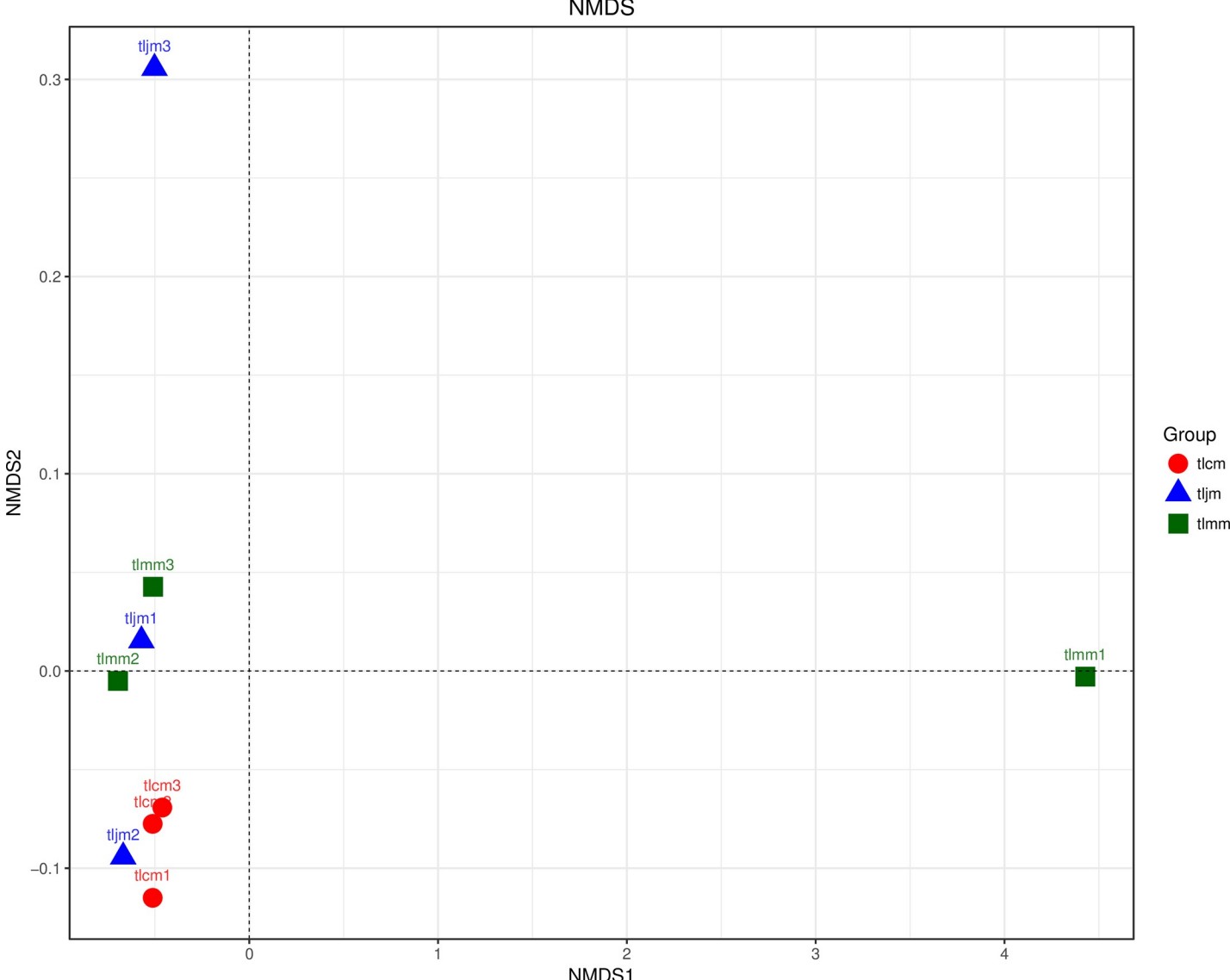

**Fig 5. NMDS analysis of the bacterial lactase gene from intestinal mucosa.** Different color points represent samples from different groups. The closer distance between the two points represented the smaller the community difference of the lactase-produced bacteria between the two samples. tlcm, tlmm and tljm represent the healthy control group, the diarrhea model group and the *D. hansenii* treatment group respectively.

*Clostridium* were only detected in the diarrhea model group. Compared with the other two groups, the abundance of lactase genes originating from *Enterococcus*, *Lysobacter* and *Salmonella* was the highest in the *D. hansenii* treatment group.

## Discussion

The gastrointestinal mucosa is the largest interface where interactions mainly occur between symbiotic bacteria and the host. A dynamic and diverse bacterial community inhabits the mucosa, which is an important part of the intestinal mucosa barrier[38]. As an important metabolic organ, intestinal flora participate in metabolism of human and animal metabolism with various microbial enzymes under gene regulation [39]. Metabolism will be affected as soon as

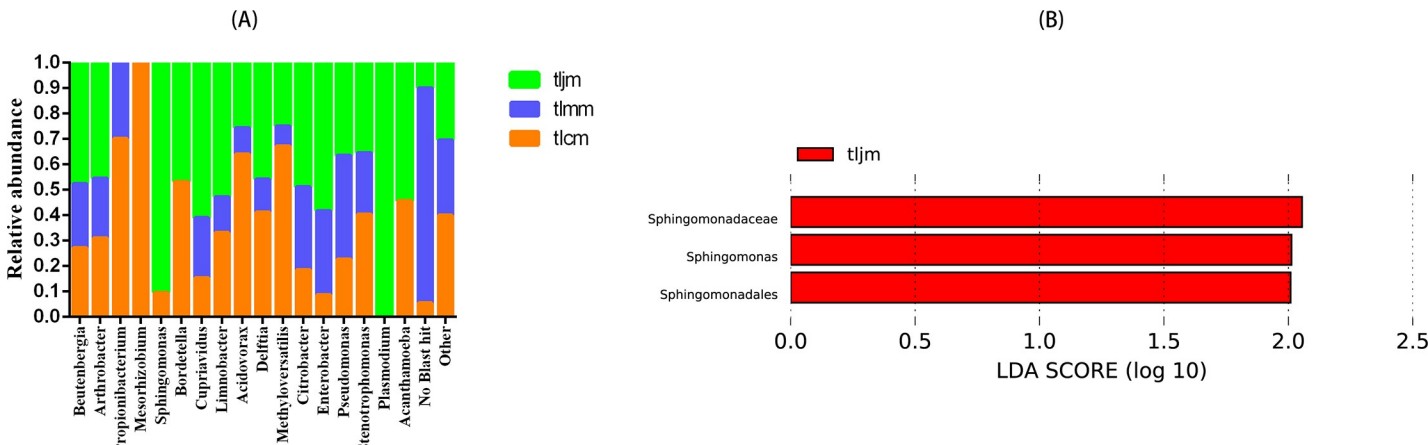

**Fig 6. (A)Relative abundance of the bacterial lactase gene from intestinal mucosa at the genus level.** tlcm, healthy control group; tlmm, diarrhea model group; tljm, *D. hansenii* treatment group. (B)**Histogram of the linear discriminant analysis**. tljm, *D. hansenii* treatment group.

the enzyme activity is absent or reduced. Furthermore, except for dysbacteriosis, the mechanisms of AAD include intestinal epithelial cilia atrophy, intestinal mucosal damage and a decrease in cellular enzyme activity as well [40]. Fortunately, AAD can be prevented through the rational use of medicines, and microecologics has become one of the common methods for treating AAD. Our previous research has proven that 25% ultramicro Qiweibaizhusan combined with 25% yeast has a relatively better therapeutic effect on AAD [24], which proves that yeast has a certain therapeutic effect on AAD. Based on the chemical nature of proteins, the activities of most enzymes are regulated by coding genes and other physicochemical factors. At the same time, with the development of modern biological technology and continuous research on intestinal microbiology, exploring the interactions between intestinal flora and intestinal diseases from the perspective of bacterial functional enzyme genes has become a new approach to conduct intestinal microbial research. Therefore, the current study was conducted to investigate the effect of *D. hansenii* on the bacterial lactase gene diversity from the intestinal mucosa of AAD mice. The results can be analyzed from three aspects: the variety of lactase-producing bacteria, the abundance of lactase-producing bacteria and the effect of *D. hansenii* on the mucosal barrier.

Intestinal bacteria mainly consist of *Proteobacteria*, *Mycobacteria*, *Actinomycetes*, *Bacteroidetes*, *Mycobacteria*, *Cyanobacteria* and *Fusobacteria*, especially *Bacteroidetes* and *Firmicutes*, which account for approximately 90% of the total[41]. Our results showed that the lactase gene in intestinal mucosa originated from *Actinobacteria*, *Firmicutes* and *Proteobacteria*, among which *Proteobacteria* accounted for more than 90%. There are differences in lactase activities produced by different bacteria in the intestine; therefore, bacterial lactase gene diversity could be reflected through differences in the population and variety of lactase-producing bacteria. NMDS analysis showed that antibiotics caused a large difference in the community structure of lactase-producing bacteria, whereas the recovery effect of *D. hansenii* treatment on the community structure was not obvious. The statistical analysis results of lactase-producing bacteria at different classification levels indicated that the *D. hansenii* had no significant effect on the population of lactase-producing bacteria in the intestinal mucosa at all classification levels except for the order level.

From the high-throughput sequencing results, the relative abundance was used to reflect the quantity difference in lactase-producing bacteria. Of the known genera, the abundance of the lactase gene originating from *Cupriavidus* in the healthy control group was significantly

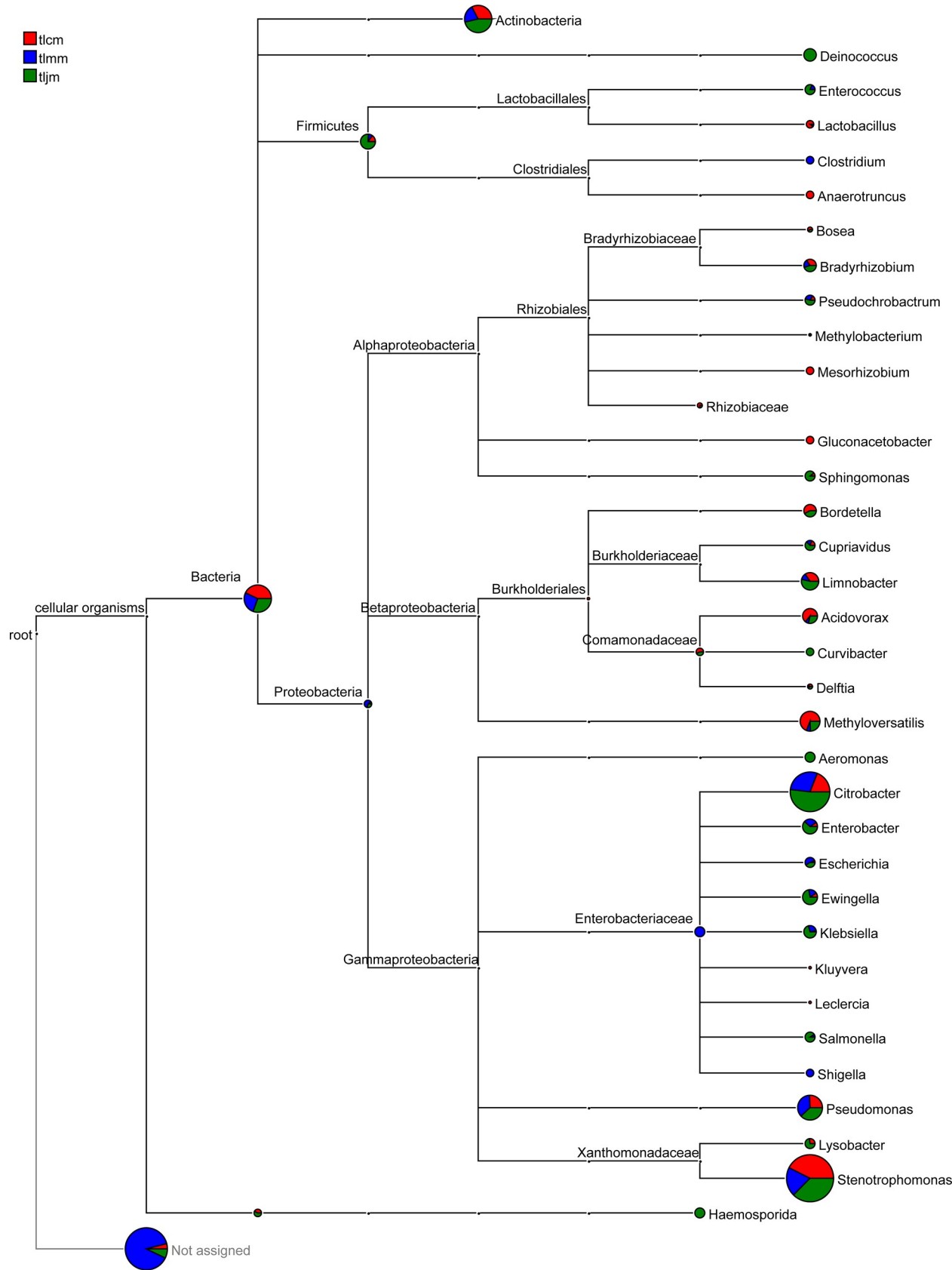

**Fig 7. Source of lactase-producing bacteria from intestinal mucosa.** The pie chart for each branch node indicated the taxon abundance in each sample;the larger the fan area, the higher the abundance of the bacterial lactase gene. tlcm, tlmm and tljm represented the healthy control group, diarrhea model group and *D. hansenii* treatment group, respectively.

higher than that in the *D. hansenii* treatment group. The study of Ueatrongchit showed that *Cupriavidus* was the genus that produced various microbial enzymes and determined that *Cupriavidus necator* produced L-threonine 3-dehydrogenase that catabolizes L-threonine[42]. *Lysobacter*, which is a genus of bacteria with fast growth, strong resistance and no pathogenicity[43], not only can secrete a variety of extracellular enzymes such as chitinase and β-1,3-glucanase, but also have significant antagonism against various pathogenic bacteria [44]. Our results showed that the lactase gene in *Deinococcus* was detected only in the *D. hansenii* treatment group and that the lactase gene in *Lysobacter* was highly abundant in the *D. hansenii* treatment group, which suggested that *D. hansenii* increased the quantity of potentially valuable lactase-producing bacteria in order to treat AAD.

The extended-spectrum cephalosporin resistant (ESC$^R$) *Enterobacteriaceae* pose a serious infection control challenge for public health [45]. The appearance of the ESC$^R$ phenotype is mainly promoted through plasmid-mediated lateral extended-spectrum β-lactamases (ESBLs) and AmpC gene transfer within *Enterobacteriaceae* [46]. The current results showed that the lactase gene in *Enterobacteriaceae* was detected only in the diarrhea model group. The Lactase gene in *Enterobacteriaceae* in the diarrhea model group was higher than that in the other groups. Moreover, the abundance of the lactase gene in *Enterococcus* was the highest in the *D. hansenii* treatment group, followed by the healthy control group and the diarrhea model group. *Enterococcus* probiotics can increase the expression of small intestinal mucosa tight junction proteins and activity of toll-like receptors 2, 4 and 9 in small intestinal mucosa, inducing an immune response in piglets [47]. All of these studies suggested that antibiotics damaged the mucosa barrier by increasing the population of bacteria with antibiotic resistance and causing dysbacteriosis. And the mucosal barrier could be restored by *D. hansenii*.

After treatment with *D. hansenii*, only a few bacterial lactase genes was recovered or increased. The recovery of lactase gene diversity in the intestinal mucosa of AAD mice was not significant. Perhaps our current results can be explained by a recent report, which determined that antibiotics had a long-term impact on intestinal microorganisms[36]. The lactase-producing bacteria in the intestinal mucosa of the diarrhea model could not return to normal levels after a few days, even following treatment with *D. hansenii*. Moreover, the activity of digestive enzymes present in the brush border of villous epithelial cells in the small intestine mucosa has the function of repairing intestinal mucosa, which is closely related to the structural integrity of the mucosa and probiotics [48,49]. In conclusion, our results suggested that antibiotics disrupted the intestinal mucosa flora in mice, and treatment with *D. hansenii* may be effective to treat diarrhea by promoting the growth of a few key lactase-producing bacteria or some beneficial bacteria to repair the intestinal mucosa structure.

## Supporting information

**S1 Dataset. The raw data of healthy control group (tlcm).**
(ZIP)

**S2 Dataset. The raw data of diarrhea model group (tlmm).**
(ZIP)

**S3 Dataset. The raw data of *D. hansenii* treatment group (tljm).**
(ZIP)

## Author Contributions

**Conceptualization:** Zhoujin Tan.

**Data curation:** Yuan Tang.

**Funding acquisition:** Wenge Li.

**Project administration:** Zhoujin Tan.

**Software:** Maijiao Peng, Guozhen Xie.

**Writing – original draft:** Yunshan He.

**Writing – review & editing:** Zhoujin Tan.

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
