## [Decision Letter · Decision Letter 0]

23 Aug 2019

PONE-D-19-19505

Influence of Debaryomyces hansenii on bacteria lactase gene diversity in intestinal mucosa of mice with antibiotic-associated diarrhea

PLOS ONE

Dear Professor Tan,

Thank you for submitting your manuscript to PLOS ONE. After careful consideration, we feel that it has merit but does not fully meet PLOS ONE’s publication criteria as it currently stands. Therefore, we invite you to submit a revised version of the manuscript that addresses the points raised during the review process.

According to the comments from the two reviewers, the manuscript has serious flaws including writing, statistical analysis, and data demonstration. 

1. You should revise your language with a native English speaker. 2. You should upload your raw data into public database such as NCBI and provide the accession number. Table 1 should be deleted as the data could be gained according to your raw data. 3. For data analysis, you should perform data normalization, i.e. each sample with same read numbers, before the subsequent data analysis.

4. Alpha diversity analysis should included richness indices and diversity indices. Please added these indices.

5. For PCoA analysis, you should include different algorithms such as weighted UniFrac, unweighted UniFrac and Bray-Curtis.

6. For differential bacteria analysis, you should perform LEfSe analysis.

We would appreciate receiving your revised manuscript by Oct 07 2019 11:59PM. To enhance the reproducibility of your results, we recommend that if applicable you deposit your laboratory protocols in protocols.io, where a protocol can be assigned its own identifier (DOI) such that it can be cited independently in the future. For instructions see: http://journals.plos.org/plosone/s/submission-guidelines#loc-laboratory-protocols

We look forward to receiving your revised manuscript.

Kind regards,

Zongxin Ling

Academic Editor

PLOS ONE

Journal Requirements:

2. Please complete and submit a copy of the ARRIVE Guidelines checklist, a document that aims to improve experimental reporting and reproducibility of animal studies for purposes of post-publication data analysis and reproducibility: https://www.nc3rs.org.uk/arrive-guidelines. Please include your completed checklist as a Supporting Information file. Note that if your paper is accepted for publication, this checklist will be published as part of your article. Specifically, please ensure that you revise your methods section to include the method of anaesthesia and euthanasia.

Additional Editor Comments (if provided):

Reviewers' comments:

Reviewer's Responses to Questions

**Comments to the Author**

1. Is the manuscript technically sound, and do the data support the conclusions?

Reviewer #1: Partly

Reviewer #2: Partly

2. Has the statistical analysis been performed appropriately and rigorously? 

Reviewer #1: No

Reviewer #2: Yes

3. Have the authors made all data underlying the findings in their manuscript fully available?

Reviewer #1: Yes

Reviewer #2: Yes

4. Is the manuscript presented in an intelligible fashion and written in standard English?

Reviewer #1: No

Reviewer #2: No

5. Review Comments to the Author

Reviewer #1: In this manuscript, the authors revealed the effects of Debaryomyces hansenii on diversity of bacterial lactase genes from intestinal mucosa of antibiotic-associated diarrhea (AAD) mice. They established diarrhea model, extracted genomic DNA of intestinal mucosa, amplified β-galactosidase gene by PCR and sequenced them using high-throughput sequencing technology. This is an interesting manuscript, generally carefully prepared, but it still requires significant revision before publication. In particular, it is lacking in essential details and information.

1.The antibiotic-associated diarrhea (AAD) mice model is the core and principal content of this manuscript. However, there is no evidence that can comfirm the model is sucsessfully established.Please provide the data.

2. As the authors mentioned “There are about 1000 species of bacteria inhabiting in human’s intestinal tract”, while they checked the β-galactosidase gene only in Lactobacillus and Escherichia coli. Why choose these two? Can the β-galactosidase genes in Lactobacillus and Escherichia coli represent that in all intestinal microorganisms?

3. Eighteen mice, which were randomly divided into three groups (6 mice per group), healthy control group, diarrhea model group and D. hansenii treatment group, were used in the present study. However, they used samples from nine mice for β-galactosidase gene amplification and sequencing. Why not check the β-galactosidase gene in all eighteen mice? Do you think the data from three sample per group is enough for clarification the influence of Debaryomyces hansenii on bacteria lactase gene diversity in intestinal mucosa of mice with antibiotic-associated diarrhea?

4. The writing need to be improved.

Reviewer #2: The submitted manuscript pertains to the influence of Debaryomyces hansenii on bacteria lactase gene diversity in mucosa of mice with antibiotics-associated diarrhea. The starting point of the study is the statement that antibiotic – associated diarrhea may be related to the decreasing activity of lactase. The aim of the work is clearly defined and has a practical perspective - to determine whether D. hansenii could be used in the treatment of antibiotic-associated diarrhea due to its effect on the regulation of lactase gene expression. The manuscript is concise, the abstract adequately describe the experimental question and results and the title is appropriate. The experimental results support the conclusions. I believe that the paper can be published once the authors address the issues below.

1) Throughout the manuscript, the use of the English language needs to be drastically improved. Numerous grammatical errors are present in the manuscript. This is a major concern that must be addressed before publication.

2) More references are appropriate, such as for the experimental techniques. For example, what is the source reference for the universal primers according to lactase gene sequences of Lactobacillus and Escherichia coli ?

3) The author should provide the data about the fecal consistency, fecal water content and stool weight after treatment with antibiotics or D. hansenii ?

6. PLOS authors have the option to publish the peer review history of their article (what does this mean?). If published, this will include your full peer review and any attached files.

Reviewer #1: No

Reviewer #2: No

---

## [Author Response · Author response to Decision Letter 0]

12 Sep 2019

Dear editor and reviewers: 

Firstly, thanks reviewers and editor for their valuable comments on our manuscript entitled " Influence of Debaryomyces hansenii on bacteria lactase gene diversity in intestinal mucosa of mice with antibiotic-associated diarrhea" (PONE-D-19-19505). Those comments are very helpful for revising and improving our paper, as well as the important guiding significance to other research. We have studied the comments carefully and made corrections which we hope meet with approval. The main corrections are in the manuscript and the response to the reviewers’ comments are as follows: 

Replies to the reviewers’ comments:

Reviewer #1: 

1.The antibiotic-associated diarrhea (AAD) mice model is the core and principal content of this manuscript. However, there is no evidence that can comfirm the model is sucsessfully established. Please provide the data.

Response: The antibiotic-associated diarrhea (AAD) first established by Ao Zeng, on the basis of which our research group has done a lot of researches, including the diversity of intestinal contents and intestinal mucosal bacteria, and the diversity of intestinal contents and intestinal mucosal lactase gene, fully demonstrating the reliability of this model.

1. Zeng A, Zhang HL, Tan ZJ, et al. The construction of mice diarrhea model due to dysbacteriosis and curative effect of ultra-micro Qiweibaizhusan. Microbiology China. 2012; 39(9): 1341-1348. https://doi.org/ 10.13344/j.microbiol.china.2012.09.012

2.Long C, Liu Y, He L, et al. Bacterial lactase genes diversity in intestinal mucosa of mice with dysbacterial diarrhea induced by antibiotics. 3 Biotech. 2018; 8(3): 176. https://doi.org/ 10.1007/s13205-018-1191-5 PMID: 29556430 

3. Long CX, He L, Guo YF, et al. Diversity of bacterial lactase genes in intestinal contents of mice with antibiotics-induced diarrhea. World Journal of Gastroenterology. 2017; 23(42):7584-7593. https://doi.org/ 10.3748/wjg.v23.i42.7584 PMID: 29204058 

2.As the authors mentioned “There are about 1000 species of bacteria inhabiting in human’s intestinal tract”, while they checked the β-galactosidase gene only in Lactobacillus and Escherichia coli. Why choose these two? Can the β-galactosidase genes in Lactobacillus and Escherichia coli represent that in all intestinal microorganisms?

Response: There are few lactase gene sequences published in NCBI database. We downloaded published 89 bacterial lactase gene sequences from NCBI database, and the sequence sources of these genes include Bifidobacterium, Lactobacillus, Enterobacter, Streptococcus, Bacillus, Micrococcus and Clostridium. According to these bacterial lactase gene sequences, we designed 7 pairs of universal primers for the analysis of bacterial lactase gene diversity. After PCR amplification and metagenomic sequencing, it was found that universal primers of bacterial lactase gene from Lactobacillus and Escherichia coli had good specificity, high sensitivity and could annotate more microbial species.

1.Long CX, He L, Liu YJ, et al. Universal primer for analysis of the diversity of intestinal bacterial lactase gene. Chinese Journal of Applied and Environmental Biology. 2017, 23(04): 758-763. https://doi.org/ 10.3724/SP.J.1145.2016.10008 

3. Eighteen mice, which were randomly divided into three groups (6 mice per group), healthy control group, diarrhea model group and D. hansenii treatment group, were used in the present study. However, they used samples from nine mice for β-galactosidase gene amplification and sequencing. Why not check the β-galactosidase gene in all eighteen mice? Do you think the data from three sample per group is enough for clarification the influence of Debaryomyces hansenii on bacteria lactase gene diversity in intestinal mucosa of mice with antibiotic-associated diarrhea?

Response: Admittedly, the bioduplication of the three samples in each group is a little low, but it can meet the minimum requirements of statistics. In our experiment, there were 6 mice in each group. When collecting intestinal mucosa, we randomly mixed the mucosa of a male and a female mouse in the group. On the one hand, the intestinal mucosa of mice was small, and we were worried that the extracted DNA was not pure enough. On the other hand, there were some differences between the intestinal flora of male mice and female mice. In order to fully reflect the characteristics of the mucosal bacteria of mice, the intestinal mucosa of a male mouse and a female mouse were mixed together.

4. The writing need to be improved.

Response: Thanks for the expert's advice. We will check the manuscript carefully to improve the readability of the article.

Reviewer #2:

1.Throughout the manuscript, the use of the English language needs to be drastically improved. Numerous grammatical errors are present in the manuscript. This is a major concern that must be addressed before publication.

Response: Thanks to the expert's valuable advice, we will ask a native English speaker to polish our manuscript.

2.More references are appropriate, such as for the experimental techniques. For example, what is the source reference for the universal primers according to lactase gene sequences of Lactobacillus and Escherichia coli ?

Response: The sources of the universal primers for the analysis of lactase gene diversity were designed by our research group. According to the lactase gene sequence registered in NCBI database, DANMAN software was used to find the conservative region of the lactase gene sequence, and the Primer Premier 5.0 software was used to design upstream and downstream primers on both ends of conserved regions. Finally, the synthesis of primers was completed by Shanghai pessenol biotechnology co., LTD. 

1.Long CX, He L, Liu YJ, et al. Universal primer for analysis of the diversity of intestinal bacterial lactase gene. Chinese Journal of Applied and Environmental Biology. 2017, 23(4): 758-763. https://doi.org/ 10.3724/SP.J.1145.2016.10008 

2.The author should provide the data about the fecal consistency, fecal water content and stool weight after treatment with antibiotics or D. hansenii ?

Response: In this study, we did not conduct quantitative analysis of fecal concentration, fecal water content and fecal weight after treatment with antibiotics or D. hansenii, but in previous articles, we have qualitatively recorded the wetness of stool after antibiotic modeling and yeast treatment in mice.

1.Guo KX, Tan ZJ, Xie MZ, et al. The synergic effect of ultra-micro powder Qiweibaizhusan combined with yeast on dysbacteriotic diarrhea mice. Chinese Journal of Applied and Environmental Biology. 2015; 21(1): 61-67. https://doi.org/ 10.3724/sp.j.1145.2013.10002 

Replies to the editors’ suggestions

1.You should revise your language with a native English speaker.

Response: Thanks for the editor's advice, in terms of language quality, we invited our colleagues working in the United States to polish the manuscript to make it more accurate.

2.You should upload your raw data into public database such as NCBI and provide the accession number. Table 1 should be deleted as the data could be gained according to your raw data.

Response: As required, we have uploaded all the original sequences to NCBI database. The accession number is SRP220043 (https://www.ncbi.nlm.nih.gov/Traces/sra_sub/), which was released on September 6, 2019. According some information on the Internet, the data uploaded to NCBI database might not be retrieved for 3 months. So I listed my NCBI database(https://www.ncbi.nlm.nih.gov/) account and password at the end. Account number: heyunshan; password: 20183371he 

3.For data analysis, you should perform data normalization, i.e. each sample with same read numbers, before the subsequent data analysis.

Response: Thanks to the editors for their advice, the sample size of each group is consistent in any one data analysis. The data analysis of this experiment was conducted by professionals of pessenol biological company, and the data were authentic and reliable with strong comparability.

4.Alpha diversity analysis should included richness indices and diversity indices. Please added these indices.

Response: Regarding Alpha diversity, Simpson index was added, which could well reflect community diversity.

5.For PCoA analysis, you should include different algorithms such as weighted UniFrac, unweighted UniFrac and Bray-Curtis.

Response: For beta analysis, PCA and NMDS analysis are selected instead of PCoA analysis. NMDS analysis is not affected by the numerical value of sample distance, and only considers the size relationship between them. For data with complex structure, NMDS analysis ranking results may be more stable.

6.For differential bacteria analysis, you should perform LEfSe analysis.

Response: In the differential bacteria analysis, we introduced LEfS analysis as required. LEfSe analysis show that Sphingomonadaceae, Sphingomonas and Sphingomonadales were the key colony members in D. hansenii treatment group, while no key colony members were found in healthy control group and diarrhea model group.

---

## [Editor Report · Decision Letter 1]

8 Oct 2019

PONE-D-19-19505R1

Influence of Debaryomyces hansenii on bacteria lactase gene diversity in intestinal mucosa of mice with antibiotic-associated diarrhea

PLOS ONE

Dear Professor Tan,

Thank you for submitting your manuscript to PLOS ONE. After careful consideration, we feel that it has merit but does not fully meet PLOS ONE’s publication criteria as it currently stands. Therefore, we invite you to submit a revised version of the manuscript that addresses the points raised as follows.

From your response, we could not find the significant improve of the manuscript especially the data analysis. You said that you had performed data normalization, which was important for the data interpretation. However, I could not find any information about the data normalization. How many reads had been normalized in your present study? from your raw data, I found that there were significantly different in the read numbers in each samples. From your Table 1 in your original submission, you could find them more clearly. From your rarefaction and ran abundance curve, I thought that you did not perform data normalization at all. I insisted that you should re-perform your data analysis before I could consider this manuscript again.

We would appreciate receiving your revised manuscript by Nov 22 2019 11:59PM. To enhance the reproducibility of your results, we recommend that if applicable you deposit your laboratory protocols in protocols.io, where a protocol can be assigned its own identifier (DOI) such that it can be cited independently in the future. For instructions see: http://journals.plos.org/plosone/s/submission-guidelines#loc-laboratory-protocols

We look forward to receiving your revised manuscript.

Kind regards,

Zongxin Ling

Academic Editor

PLOS ONE

---

## [Author Response · Author response to Decision Letter 1]

6 Nov 2019

Dear Editors:

 Due to the negligence of the company, there are some problems with the raw data of tlmm1 sample. As a result, when you carefully examine the raw data, you find that there were significantly different in the read numbers in each samples. After communication with the company, the raw data of tlmm1 sample has been recovered from the company's database. I want to upload all the original data to NCBI database again, but due to duplication with the previously uploaded data, so it has been unable to pass the review of NCBI database. Therefore, the raw data will be sent to PLOS ONE's submission system in the form of attachment. Data analysis has asked the company to analyze the original data according to the latest industry standards, and the analysis results are basically the same as before. The language of the article has been requested to be retouched by AJE company, and the retouching certificate has been uploaded in the form of attachment. Thank you again for taking time out of your busy schedule to review the manuscript.

Best regards.

Yours sincerely,

Yunshan He

Correspondence author:

Name: Wenge Li; Zhoujin Tan

E-mail: 641386565@qq.com; tanzhjin@sohu.com

---

## [Editor Report · Decision Letter 2]

13 Nov 2019

Influence of Debaryomyces hansenii on bacteria lactase gene diversity in intestinal mucosa of mice with antibiotic-associated diarrhea

PONE-D-19-19505R2

Dear Dr. Tan,

We are pleased to inform you that your manuscript has been judged scientifically suitable for publication and will be formally accepted for publication once it complies with all outstanding technical requirements.

With kind regards,

Zongxin Ling

Academic Editor

PLOS ONE
---

## [Editor Report · Acceptance letter]

20 Nov 2019

PONE-D-19-19505R2 

Influence of *Debaryomyces hansenii* on bacterial lactase gene diversity in intestinal mucosa of mice with antibiotic-associated diarrhea 

Dear Dr. Tan:

I am pleased to inform you that your manuscript has been deemed suitable for publication in PLOS ONE. Congratulations! Your manuscript is now with our production department. 

With kind regards,

on behalf of

Dr. Zongxin Ling 

Academic Editor

PLOS ONE